

# In silico analysis on the functional and structural impact of Rad50 mutations involved in DNA strand break repair

Juwairiah Remali[1], Wan Mohd Aizat[2], Chyan Leong Ng[2],
Yi Chieh Lim[3], Zeti-Azura Mohamed-Hussein[2,4] and Shazrul Fazry[1,5]

[1] Department of Food Sciences, Faculty of Science and Technology, Universiti Kebangsaan Malaysia, Bangi, Selangor, Malaysia
[2] Institute of Systems Biology (INBIOSIS), Universiti Kebangsaan Malaysia, Bangi, Selangor, Malaysia
[3] Danish Cancer Society, Research Centre Strand Boulevard, Copenhagen, Denmark
[4] Department of Applied Physics, Faculty of Science and Technology, Universiti Kebangsaan Malaysia, Bangi, Selangor, Malaysia
[5] Pusat Penyelidikan Tasik Chini, Fakulti Sains dan Teknologi, Universiti Kebangsaan Malaysia, Bangi, Selangor, Malaysia

Corresponding author
Shazrul Fazry, shazrul@ukm.edu.my

## ABSTRACT

**Background:** DNA double strand break repair is important to preserve the fidelity of our genetic makeup after DNA damage. Rad50 is one of the components in MRN complex important for DNA repair mechanism. Rad50 mutations can lead to microcephaly, mental retardation and growth retardation in human. However, Rad50 mutations in human and other organisms have never been gathered and heuristically compared for their deleterious effects. It is important to assess the conserved region in Rad50 and its homolog to identify vital mutations that can affect functions of the protein.

**Method:** In this study, Rad50 mutations were retrieved from SNPeffect 4.0 database and literature. Each of the mutations was analyzed using various bioinformatic analyses such as PredictSNP, MutPred, SNPeffect 4.0, I-Mutant and MuPro to identify its impact on molecular mechanism, biological function and protein stability, respectively.

**Results:** We identified 103 mostly occurred mutations in the Rad50 protein domains and motifs, which only 42 mutations were classified as most deleterious. These mutations are mainly situated at the specific motifs such as Walker A, Q-loop, Walker B, D-loop and signature motif of the Rad50 protein. Some of these mutations were predicted to negatively affect several important functional sites that play important roles in DNA repair mechanism and cell cycle signaling pathway, highlighting Rad50 crucial role in this process. Interestingly, mutations located at non-conserved regions were predicted to have neutral/non-damaging effects, in contrast with previous experimental studies that showed deleterious effects. This suggests that software used in this study may have limitations in predicting mutations in non-conserved regions, implying further improvement in their algorithm is needed. In conclusion, this study reveals the priority of acid substitution associated with the genetic disorders. This finding highlights the vital

roles of certain residues such as K42E, C681A/S, CC684R/S, S1202R, E1232Q and D1238N/A located in Rad50 conserved regions, which can be considered for a more targeted future studies.

## INTRODUCTION

The DNA repair process exists in all organisms including both prokaryotes and eukaryotes, and most of the related proteins in this process are known to be highly conserved throughout biological evolution. One such protein complex involved in eukaryotic DNA repair process is MRN complex, and it is comprised of three proteins: meiotic recombination 11 (Mre11), DNA repair protein Rad50, and nibrin (called Nbn or Nbs1). These proteins play an important role in maintaining the genomic integrity by orchestrating DNA damage checkpoint, telomere maintenance, homologous recombination (HR) as well as non-homologous end joining repair (NHEJ) mechanism (*Van den Bosch, Bree & Lowndes, 2003*). MRN complex is one of the first factors to be localized to DNA lesions where it has a structural role by tethering and stabilizing broken chromosomes (*De Jager et al., 2001*; *Van den Bosch, Bree & Lowndes, 2003*).

Null mutations in MRN complex have been shown to be lethal in higher eukaryotes such as in embryonic stem cells (*Luo et al., 1999*). In addition, mutations in the *Nbs1* gene, can cause Nijmegen breakage syndrome (NBS), whereas Mre11 mutations resulted in Ataxia telangiectasia-like disease syndrome (ATLD) (*Carney et al., 1998*). So far, studies of Nbs1 and Mre11 deficiencies in human have been extensively investigated through cells and clinical data obtained from NBS and ATLD patients (*Barbi et al., 1991*; *Waltes et al., 2009*). Unfortunately, investigation of the effect of Rad50 mutations on human is very limited due to the fact that only one patient with fully characterized Rad50 deficiency (known as NBS like disorder (NBSLD)) has been reported (*Waltes et al., 2009*). This NBSLD patient, with microcephaly, bird-like features, radiosensitivity and delayed development, was revealed to have inherited heterozygous mutations from her parents (*Barbi et al., 1991*). The first mutation (c.3277C/T; p.R1093X) on exon 21 was maternally inherited causing a premature termination codon, thus producing a truncated Rad50 protein, whereas the second mutation on the exon 25 (c.3939A/T) was paternally inherited and it has changed the stop codon of normal Rad50 to a tyrosine codon, thereby producing a larger Rad50 protein (*Waltes et al., 2009*). Both mutations interestingly give rise to the hypomorphic characterization of the Rad50 expressions in this patient (*Gatei et al., 2011*). The cause of this characteristic is still being debated to this day. Given that perturbation of Rad50 structure and function could contribute to genomic instability (*Assenmacher & Hopfner, 2004*), it is therefore important to decipher its conserved domains and genetic polymorphism.

Single nucleotide polymorphism (SNP) is one of the most common types of genetic variation in human (*Lee et al., 2005*). Even though most of the polymorphic changes do not

affect normal cellular function, some variants do influence gene expression or translated protein function (*Risch & Merikangas, 1996*; *Collins, Guyer & Charkravarti, 1997*). For instance, cystic fibrosis (*Bartoszewski et al., 2010*), sickle-cell anemia (*Shaikho et al., 2017*), and β-thalassemia (*Traeger et al., 1980*) are examples of diseases resulted from SNPs. Nearly half of the disease-related mutations are derived from nonsynonymous SNPs (nsSNPs), a single base change that alters the amino acid sequence of the encoded protein (*Cargill et al., 1999*; *Halushka et al., 1999*). Although it is remarkably important to reveal the connection between SNPs and related diseases, the accelerating number of known SNPs have made it very difficult to discriminate between pathogenic and neutral variants through experimental validations (*Tranchevent et al., 2011*). Therefore, bioinformatic prediction tools have become extremely critical for the initial analysis of their molecular functions as well as prioritization of further experimental characterization including deciphering the effects of Rad50 SNPs (*Bendl et al., 2014*). Furthermore, prioritization of disease candidates genes from experiment and databases evidence is essential for further pathological investigation (*Piro & Di Cunto, 2012*). Several investigations on Rad50 mutations have been reported in human (*Waltes et al., 2009*; *Gatei et al., 2011*), mice (*Bender et al., 2002*; *Roset et al., 2014*), yeast (*Alani, Padmore & Kleckner, 1990*; *Chen et al., 2005*), and archaea (*Koroleva et al., 2007*) yet there are still no reports that compare these experimental results with in silico prediction, which will be important for the protein functional annotation. Moreover, a number of different SNPs for Rad50 have been deposited in SNP databases but their impact on the cellular regulation have not been thoroughly investigated thus far.

Hence, the aim of this study was to identify the functional and structural effects of amino acid mutations in Rad50 gathered from exhaustive literature review and SNP database (SNPeffect 4.0) search. Rad50 sequences in different organisms including human and selected animals (chimpanzee, rats, mice, zebra fish, rabbit and fruit fly) and yeasts were compared and aligned to identify their conserved residues. Mutations that contributed to the most damaging effects were then analyzed in silico using PredictSNP for the amino acid impact after the substitution, MutPred for predicting molecular mechanism, SNPeffect for identification of protein or amyloid aggregation as well as I-Mutant and MuPro for protein stability after the mutation. Such approach was also successfully reported by several researchers studying the impact of various SNPs. For example, *Marín-Martín et al. (2014)* studied the impact of SNPs in the ABCA1 transporter gene by cross validating their prediction with experimentally reported data. Another study by *Fawzy et al. (2015)* also validated their in silico approach finding by means of comparison with available literature to study gene polymorphisms in obese children and adolescents. In this study, Rad50 mutations gathered from various studies are compared with their in silico predictions. This is highly valuable in understanding Rad50 functional roles especially during DNA strand break, allowing prioritization of mutations or sites to be studied in future in vivo studies, whilst bearing in mind its possible impact on human. Ultimately, this may help on the development of precision medicine for Rad50 mutations in humans.

## MATERIALS AND METHODS

### Multiple sequence alignment analysis and conserved domain analyses

Human Rad50 protein sequence was obtained from National Center for Biotechnology Information (NCBI). The sequence similarity search tool, BLASTP from the NCBI server (http://blast.ncbi.nlm.nih.gov/Blast.cgi) was used to find homologs for Rad50. To investigate the similarity between Rad50 protein in human and other organisms such as *Danio rerio, Mus musculus, Rattus norvegicus, Pan troglodytes, Oryctolagus cuniculus, Drosophila melanogaster, Saccharomyces cerevisiae* and *Schizosaccharomyces pombe*, a multiple sequence analysis (MSA) was conducted using Clustal Omega (https://www.ebi.ac.uk/Tools/msa/clustalo/) with default settings to determine consensus and conserved regions between the multiple sequences of different organisms (*Sievers & Higgins, 2018*). Meanwhile, InterPro (http://www.ebi.ac.uk/interpro/) was used to identify the domains and motifs using human sequence (*Finn et al., 2017*). InterPro results are classified into several types (families, domains, motif or sites) depending on the biological entity they represent (*Finn et al., 2017*). Using this tool, Rad50 protein sequence was classified into families and the presence of domains and important sites were predicted. ClustalX software (*Thompson, Gibson & Higgins, 2002*) was used to view and analyze the conserved regions within the domains and motifs in the selected proteins.

### Data mining of Rad50 mutation from literature and SNPs database

Rad50 mutations were identified from previous published manuscripts using PubMed database and their functional impacts were extracted for comparison. Besides that, naturally occurring single nucleotide polymorphisms (SNPs) in Rad50 were retrieved from SNPeffect 4.0 database (http://snpeffect.switchlab.org/about) (*De Baets et al., 2012*) (date of access: 7 April 2018). SNPeffect 4.0 database currently contains more than 60,000 human SNPs gathered from human avariance list available at UniProt website (https://www.uniprot.org/). It specifically focuses on the molecular characterization, annotation of diseases as well as polymorphism variants in human proteins (*De Baets et al., 2012*). All these available Rad50 protein mutations (obtained from both literature and databases) have been aligned using pairwise alignment through Clustal Omega between human sequence and other organisms' sequence, individually. From this analysis, we identified similar mutation sites in human. All the identified equivalent mutations in human were manually refined, for example removing the same residues and mutations that has been studied by several different researchers to identify the non-redundant mutations (Table S1). Identified mutations (after converting to equivalent residues in human) were then mapped into Fig. S1.

### Secondary structure prediction and analysis of 3D modeling

The Rad50 templates identified from the BLAST analysis also were used to develop secondary structure and 3D model. The PSIPRED program (http://bioinf.cs.ucl.ac.uk/psipred/) has been utilized for secondary protein structure prediction (*Buchan et al., 2013*). Secondary structure prediction has revealed a clear distribution of alpha helix, beta sheet and coil in *H. sapiens* (Helix: 74.69%, coil; 18.29 and beta sheet; 7.01%) (Fig. S2).

Databases such as UniProt (https://www.uniprot.org/) and Protein Data Bank (PDB) (https://www.rcsb.org/) were used to identify structural information regarding Rad50 protein in human. Rad50 protein sequence also has been BLAST searched against Protein Data Bank (PDB) sequence in Network Protein Sequence @nalysis (NPS@) (https://npsa-prabi.ibcp.fr/) to identify the most identical structure. The incomplete structure has been further predicted using fold recognition method using Protein Homology/analogY Recognition Engine Version 2.0 (Phyre2) (http://www.sbg.bio.ic.ac. uk/phyre2) (*Kelley et al., 2015*). Phyre2 is an online tool to predict and analyze protein structure, function and mutations which uses advanced remote homology detection methods to build 3D models, predict ligand binding sites and analyze the effect of amino acid variants (e.g., nonsynonymous SNPs (nsSNPs)) for a protein sequence (*Kelley et al., 2015*). Rad50 sequence was submitted to the webserver to interpret the secondary and tertiary structures of the model, domain composition and quality. 3D model of Rad50 was run under 'intensive' mode that generates a complete full-length model of a protein sequence by using multiple template modeling and simplified ab initio folding simulation (*Kelley et al., 2015*). UCSF Chimera software was used to view and to analyze the 3D structure (*Pettersen et al., 2004*).

## Prediction of deleterious effects of Rad50 mutations using in silico tools

The Rad50 mutations were in silico predicted using PredictSNP to determine their possible molecular impacts in human (https://loschmidt.chemi.muni.cz/predictsnp1/) (*Bendl et al., 2014*). Its benchmark dataset contains over 43,000 mutations obtained from the Protein Mutant Database and the UniProt database (*Bendl et al., 2014*). This tool incorporated six established prediction tools; such as Multivariate Analysis of Protein Polymorphism (MAPP) (*Stone & Sidow, 2005*), Predictor of human Deleterious Single Nucleotide Polymorphisms (PhD-SNP) (*Capriotti & Fariselli, 2017*), PolyPhen-1 (*Ramensky, 2002*), PolyPhen-2 (*Adzhubei, Jordan & Sunyaev, 2013*), Sorting Intolerant from Tolerant (SIFT) (*Sim et al., 2012*) and Single-Nucleotide Amplified Polymorphisms (SNAP) (*Bromberg & Rost, 2007*) to provide a more accurate and robust comparison. We classified the mutations as deleterious if five to seven analyses performed were identified as damaging in PredictSNP. For instance, an in silico prediction was considered accurate when a given mutation predicted to be deleterious (as performed in this study) was also found experimentally deleterious (either in vitro or in vivo with phenotypes such as embryonic lethality, growth defect and/or cancer predisposition) based on previous cited studies. Conversely, the prediction is inaccurate if such deleterious mutations was predicted as neutral or tolerant.

## Molecular mechanism of amino acid substitutions

To determine the molecular mechanism based on pathogenicity of amino acid substitutions in Rad50, MutPred2 (*Pejaver et al., 2017*) (http://mutpred2.mutdb.org/index. html) analysis was carried out. This program predicts the pathogenicity and molecular

impacts of amino acid substitutions potentially affecting the phenotype. It is trained on a set of 53,180 pathogenic and 206,946 unlabeled (putatively neutral) variants obtained from the Human Gene Mutation Database (HGMD) (*Stenson et al., 2017*), SwissVar (*Mottaz et al., 2010*), dbSNP (*Sherry et al., 2001*) and inter-species pairwise alignment (*Pejaver et al., 2017*). The output of MutPred contains a general probability that the amino acid substitution is deleterious/disease-associated, and a list of rank of specific molecular alterations potentially affecting the phenotype with its *p*-value (<0.05).

## Prediction of molecular and structural effects of protein coding variants in Rad50 mutation

Prediction of molecular and structural effects of protein coding variants in Rad50 mutations was performed using SNPeffect4.0 (*De Baets et al., 2012*) (http://snpeffect. switchlab.org/about). The analysis includes predictions of the aggregation prone regions in a protein sequence (TANGO), amyloid-forming regions (WALTZ) and chaperone binding site (LIMBO). The range of prediction score differences outside −50 to 50 for mutants are considered significant (*De Baets et al., 2012*). SNPeffect also uses FoldX (*Schymkowitz et al., 2005*) to analyze the effect of mutations on the structural stability. However, as structure quality is important for the accuracy of delta G predictions for stability, model structures with less than 90% sequence identity to the modeling template structure will not be modeled (*De Baets et al., 2012*).

## Analysis of protein stability

The stability of Rad50 upon single amino acid residue mutations were predicted using MUpro (http://mupro.proteomics.ics.uci.edu/) (*Cheng, Randall & Baldi, 2006*) and I-Mutant 3.0 (http://gpcr2.biocomp.unibo.it/cgi/predictors/I-Mutant3.0/I-Mutant3.0.cgi) (*Capriotti, Fariselli & Casadio, 2005*) using default setting, for instance temperature was set at 25 °C and pH 7. Mu-Pro and I-Mutant 3.0 are valuable tools for protein stability prediction and analysis, even when the protein structure is not yet known with atomic resolution. Both use support vector machines (SVM)-based tool to predict protein stability changes for single amino acid mutations either from both sequence or structural information which correctly predicts with over 80% accuracy using cross validation methods (datasets and experimental) (*Capriotti, Fariselli & Casadio, 2005*; *Cheng, Randall & Baldi, 2006*). Rad50 protein sequence was searched against the web server and energy changes (ΔΔG) were recorded. Negative value for ΔΔG represents a decrease in protein stability whereas positive value for ΔΔG represents an increase in stability.

## RESULTS

### Rad50 data acquisition and MSA analysis

Human Rad50 sequence from NCBI database contains 1312aa with the accession number of AAB07119.1. Sequence homology search of the human Rad50 protein was performed against NCBI nonredundant protein databases (*E*-value ≤ 1E−05) and the result was downloaded for further analysis. Out of 500 sequences, six sequences were choosen for

MSA analysis from diverse organisms such as *D. rerio*, *M. musculus*, *R. norvegicus*, *P. troglodytes*, *O. cuniculus*, and *D. melanogaster*. Two sequences, *Saccharomyces cerevisiae* and *Schizosaccharomyces pombe* were also included due to widely being used as models in previous Rad50 studies (Table S1).

## Analysis of protein domains

Domain identification analysis showed that Rad50 contains three P-loop containing nucleoside triphosphate hydrolase (P-loop NTPase) domains which belong to ATP Binding Cassette (ABC) protein superfamily (*De La Rosa & Nelson, 2011*). It is located near the N- and C-terminal, at the residue number of 25-103, 130-227 and 1196-1279 (Fig. 1A). Residue annotation showed that Rad50 has six specific motifs including Walker A and Q-loop that are located at the N-terminal whereas Rad50 signature motif, Walker B, D-loop and H-loop/switch region are located at C terminal (Fig. 1A) (*De La Rosa & Nelson, 2011*). It also has a domain called zinc hook (635-734aa) located at C-terminal region (Fig. 1A) (*Hopfner et al., 2002*). Multiple sequence alignment (MSA) analysis between human Rad50 and its homologous genes (*D. rerio*, *M. musculus*, *R. norvegicus*, *P. troglodytes*, *O. cuniculus*, *D. melanogaster*, *S. cerevisiae* and *S. pombe*) also revealed that these specific motifs are highly conserved (Fig. 1B).

## Mutation datasets from the literature and database searches

In order to identify the Rad50 mutations, literature pertaining to the topic was exhaustively searched and 18 articles over the period of 1990 to 2017 were identified. All these mutations from different organisms were listed in Table S1. There are 103 mutations identified which mostly occurred in the protein domains and motifs with various biological effects (Table S2). In order to obtain equivalent mutations in human, pairwise alignment was performed individually between each organism (*D. rerio*, *M. musculus*, *R. norvegicus*, *P. troglodytes*, *O. cuniculus*, *D. melanogaster*, *S. cerevisiae and S. pombe*) and the Rad50 human sequence as a reference (Table S1). Then, MSA analysis was carried out between these sequences from different organisms (including human) to identify consensus regions (Fig. 1; Fig. S1). Further refinement such as integrating similar mutations that occurred at the same positions (for examples; S1202R, K42R, S679R, P682E, V683R, R1214E, K6E, and R81I) (Table S2) from different organisms of which a total of 80 different mutations or non-redundant mutation were identified. All these mutations have been mapped based on equivalent residues in human (Fig. S1). From SNPeffect 4.0 database, another 13 SNP mutations were also identified (Table S3). However, from the total of 103 mutations obtained from literature, only 42 residues of the Rad50 protein mutations were known to contribute to the most damaging effects in vitro and in vivo such as embryonic lethality (*Bender et al., 2002*; *Roset et al., 2014*) and growth defect (Table 1; Table S2) (*Alani, Padmore & Kleckner, 1990*; *Bhaskara et al., 2007*; *Waltes et al., 2009*; *He et al., 2012*; *Barfoot et al., 2015*; *Hohl et al., 2015*). Most of these deleterious mutations reside at the specific motifs such as Walker A, Q-loop, zinc hook, Rad50 signature motif, Walker B and D-loop (Fig. 1B) that become our primary research focus (Fig. 1B).

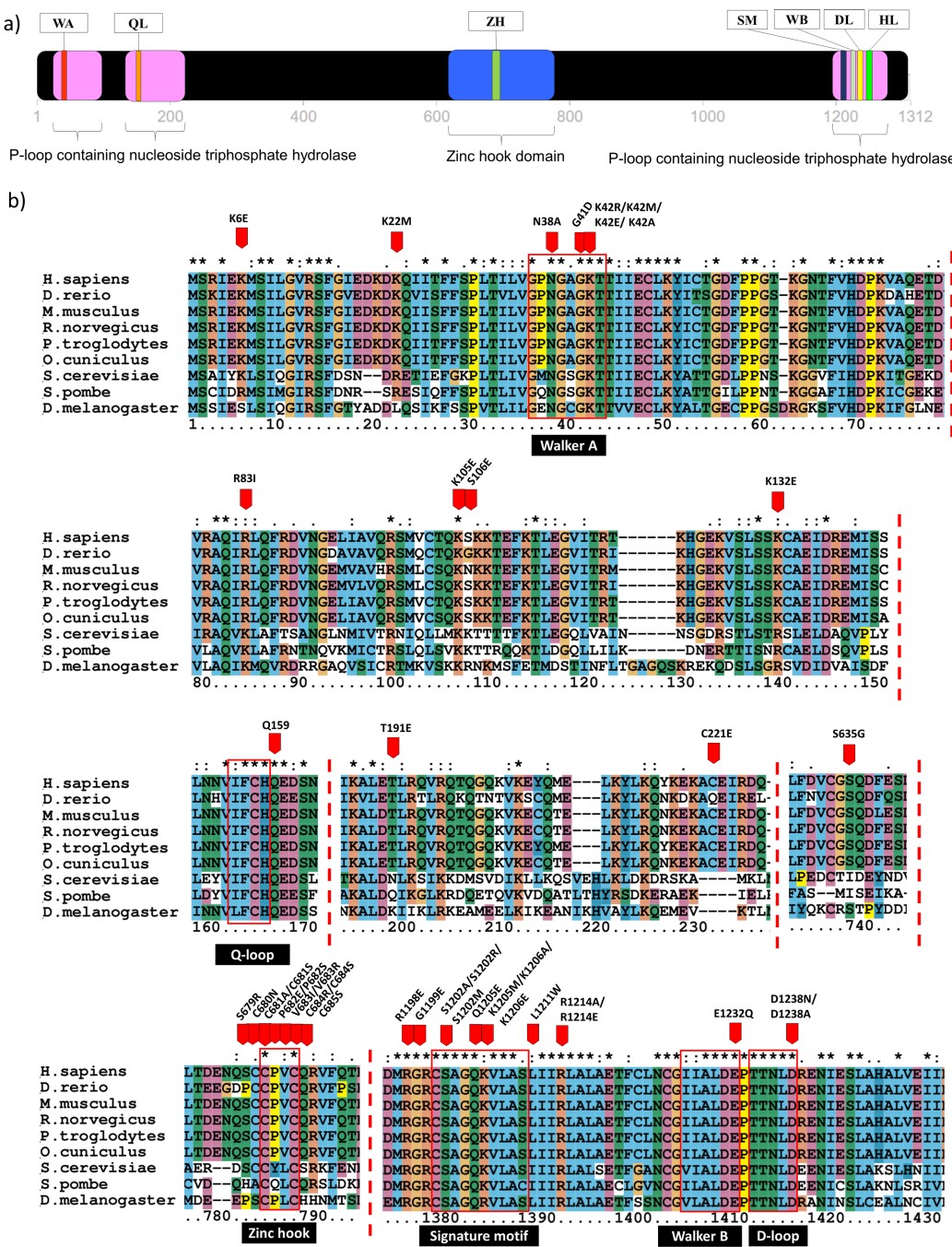

**Figure 1 Domain analysis and multiple sequence alignment.** Domain analysis using InterPro shows that Rad50 contains P-loop containing nucleoside triphosphate hydrolase domain belongs to the ATP Binding Cassette (ABC) protein superfamily (pink box) as well as a special domain called zinc hook, which so far does not overlap with any homologous superfamilies (blue box) (a). ABC protein consists of six conserved motifs; that is, Walker A (WA), Q-loop (QL), signature motif (SM), Walker B (WB), D-loop (DL), and H-loop (HL) which make up the nucleotide binding domain. Zinc hook domain contains a conserved CxxC motif located at the residue number 681–684 (A). All deleterious residues identified from the literature were highlighted based on human equivalent mutation (Table S1) and those occurring only in the conserved regions are shown in (B). Multiple sequence alignment (MSA) analysis of Rad50 sequences dataset (human, *D. rerio* (zebrafish), *M. musculus* (mouse), *R. norvegicus* (rat), *P. troglodytes* (chimpanzee), *O. cuniculus* (rabbit), *D. melanogaster* (fruit fly), *S. cerevisiae* (yeast) and *S. pombe*

**Figure 1 (continued)**
(yeast)) showed conserved residues in specific motifs (B). An "*" (asterisk) indicates position which has a single, fully conserved residue. A ":" (colon) indicates conservation between groups of strongly similar properties—scoring > 0.5 in the Gonnet PAM250 matrix. A "." (period) indicates conservation between groups of weakly similar properties—scoring ≤ 0.5 in the Gonnet PAM250 matrix.

## 3D structure modeling of Rad50

Currently there is no complete structure of human Rad50 available. Nonetheless, a crystal structure of Rad50 hook and coil–coil domain (HCC) that contains 182 residues (residue 585-766) has been determined (PDB ID: 5GOX) (*Park et al., 2017*), which represents 13% of the Rad50 structure in human. We attempted to predict a more complete human Rad50 3D structure model using homology modeling. Homology modeling program Phyre2 successfully predict the N-terminal of 276 residues (2-278) and C-terminal of 155 residues (1153-1306) of human Rad50 with 100% confidence level using the template model (PDB ID: 5DAC) from *Chaetomium thermophilum* (*Seifert, Lammens & Hopfner, 2015*) that share 66% sequence identity (Fig. S2). As a result, half of human Rad50 protein structure was obtained. The regions with no 3D structure information available are residues 279-584 and residues 767-1152 (Fig. S2), which were mainly predicted to consist of alpha helices secondary structure (Fig. S2). Pairwise alignment of Rad50 sequence between *C. thermophilum* (1315aa) and human (1312aa) showed about 30% sequence identity (Fig. S2). The result showed that the partial structure of *C. thermophilum* that has been determined (black line) are highly conserved with the human sequence (Fig. S2) suggesting that the human structure should also share high structure similarity to *C. thermophilum* at these regions. In agreement to this, results from Phyre2 prediction showed that the N-terminal and C-terminal of Rad50 form a globular and coil–coil domain, similar to the structure of *C. thermophilum* (Fig. 2A). With the generated model, six motifs of Rad50 namely Q-loop, Walker B, signature motif, D-loop, Walker A and H-loop were identified and marked in the 3D structure (Fig. 2A). All identified deleterious residues found in the domain were also marked as shown in Fig. 2B. To correlate the deleterious residues in the Rad50 HCC domain with zinc hook motif that was not found in the model, the structure of 181 residues (residue 585-766) that has been determined (PDB ID: 5GOX) independently was employed (Figs. 2C and 2D) for functional analysis.

## Analyses of Rad50 mutation deleterious effects

All of these 42 mutations (based on mutations from other organisms mapped to human) (Figs. 1B and 2B) were then analyzed using bioinformatics analyses such as impact of amino acid substitutions (PredictSNP), molecular mechanism (MutPred), structural phenotyping (protein and amyloid aggregation) (SNPeffect 4.0) and protein stability (MuPro and I-Mutant 3.0) (Table 2). All raw data from each analysis has been supplied as Supplemental Data (Table S3 for PredictSNP analysis, Table S4 for MutPred analysis, Table S5 for SNPeffect analysis and finally Table S6 for I-Mutant and MuPro analysis). The results showed that most of the deleterious effects fall into specific motifs such as

**Table 1 Summary of the most damaging effects of Rad50 mutations obtained from previous in vitro and in vivo experiments.**

| Motif/domain | Mutations | Organism | Effects | References |
|---|---|---|---|---|
| Walker A | K40A/R/E | S. cerevisiae | • HR and NHEJ defects and lower ATPase activity | Chen et al. (2005) |
| Walker A D-loop | N38A, D512N/A | T4 bacteriophage | • Naturally occurring mutation of CFTR protein | De La Rosa & Nelson (2011) |
| | | | • Reduce in ATP activity | |
| ATP binding domain and Walker A | G39D, K40E, K81I, R20M | S. cerevisiae | • Total defect in formation of viable spore | Alani, Padmore & Kleckner (1990) |
| ATP binding domain | K6E, K22M, R83I | M. musculus | • Embryonic lethality, growth defect, cancer predisposition, hematopoietic and spermatogenic depletion | Bender et al. (2002) |
| Walker A | K39R, K42M | D. radiodurans | • Prevented ATP binding and hydrolysis | Koroleva et al. (2007) |
| ATPase binding domain, Walker B and Signature motif | K115E, K175E, K182E, R94E, K95E, R765E | T. maritima | In vitro: Thermotoga maritima<br>• K175E, K182E, K115E Reduced DNA binding<br>• R94E and K95E: Important for DNA binding<br>• R765E: Diminished DNA binding<br>• E798Q: Low affinity to DNA<br>• S768R: Reduced DNA binding | Rojowska et al. (2014) |
| | E798Q, S768R, K103E, K104E, R131E, R1202E, S1205R, E1235Q | S. cerevisiae | In vivo: Saccharomyces cerevisiae<br>• S1205R and E1235Q double mutation: Unable to rescue the impaired DNA damage response<br>• K103E, K104E and R131E: Strongly affected DNA binding and moderate reduction in telomere length<br>• K103E and R131R (double mutation) and R1201E: Significantly reduced telomere length<br>• S1205R: Significantly reduced telomere length | |
| Zinc hook | S679R, P682E, V683R | M. musculus | • Lethality in mice. Hydrocephalus, defects in primitive hematopoietic and gametogenic cells | Roset et al. (2014) |
| | C684N, C685A, P686A, V6871, C688R, Q689S | S. cerevisiae | • Defective to be recruited to chromosomal double strand break<br>• Phenotype as severe as Rad50 null mutant<br>• Defective in ATM activation, HR, sensitive to irradiation and ATR activation | He et al. (2012) |
| | C288S, C291S | T4 bacteriophage | • Double mutation is lethal | Barfoot et al. (2015) |
| | S635G | H. sapiens | • Chromosomal instability<br>• Defective ATM-dependent signaling | Gatei et al. (2011) |
| | S685R, Y688E, L689R | S. cerevisiae | • S685R and Y688E double mutation: Sporulation efficiency and viability were severely impaired followed by L689R<br>• Rad50-Mre11 interaction was strongly impaired, partial suppression of telomere and meiotic defects | Hohl et al. (2015) |
| Rad50 Signature motif | R805E, L802W | P. furiosus | • L802W: Decrease dimerization in ATP, hydrolysis and cleavage site<br>• R805E: Poorly grown in camptothecin; inability to repair endogenous DNA damage by HR and showed defect in resection in HO endonuclease induced | Deshpande et al. (2014) |
| | K1187A, K1187E, R1195A, R1195E | S. pombe | • K1187A: Sensitive in higher dose of clastogens<br>• K1187E, R1195A and R1195E: Significantly sensitive to clastogen agents and were deleterious as Rad50 null mutation | Williams et al. (2011) |

| Motif/domain | Mutations | Organism | Effects | References |
|---|---|---|---|---|
| | S471A/R/M, E474Q, K475M | T4 bacteriophage | • S471A/R.M, E474Q and K475M: Residues involved in the allosteric transmission between DNA and ATP binding sites | Herdendorf & Nelson (2011) |
| | S1205R | S. cerevisiae | • S1202R: Reduced adenylate kinase | Bhaskara et al. (2007) |
| | S793R | P. furiosus | • S793R: Deficient in ATP-dependent dimer formation and ATP binding | |
| | S1202R | H. sapiens | • S1202R and S1205R: Low level of adenylate kinase | |
| | | | • S1205R: Telomere shortening, not support spore viability | |
| Signature motif and Q loop | S793R, Q140H | P. furiosus | • S793R: Analogs to the mutation in CFTR (S549R) gene that results cystic fibrosis | Moncalian et al. (2004) |
| | | | • S793R: Prevented ATP binding | |
| | S1205R | S. cerevisiae | • S1205R S. cerevisiae: Failed to complement Rad50 deletion strain in DNA repair assay | |
| | | | • S783R and Q140H: Halted ATP-dependent activities | |
| ATPase domain | R1093 (stop) c.3939A/T | H. sapiens | • Nijmegen breakage syndrome like disorder (NBSLD) | Waltes et al. (2009) |

**Note:**
HR, homologous recombination; NHEJ, non-homologous end joining repair; CFTR, cystic fibrosis transmembrane conductance regulator; ATP, adenosine tri-phosphate; ATM, ataxia-telangiectasia mutated; ATR, ATM-and Rad3-Related. Refer to Table S2 for the description of all mutations.

Walker A, Q-loop, Rad50 signature motif, Walker B and D-loop (Table 2; Fig. 2A). Previous analysis also revealed that mutations at these motifs contributed to a number of biological defects such as growth defect (Alani, Padmore & Kleckner, 1990; He et al., 2012; Hohl et al., 2015) embryonic lethality (Bender et al., 2002; Roset et al., 2014), cancer predisposition (Bender et al., 2002; Roset et al., 2014), hematopoietic and spermatogenic depletion (Bender et al., 2002; Roset et al., 2014) (Table 1). Several mutations at the zinc hook region (C681A, C681S, P682E, C684R and C684S) and ATPase/coiled-coil domain (K6E and K132E) also showed to be deleterious (Table 2; Figs. 2C and 2D).

Furthermore, mutations located at Walker A (Fig. 2A) were predicted to affect catalytic and allosteric site, loss or gain of methylation, alteration of DNA binding, metal binding, ordered interface and the loss of relative solvent accessibility, which all are depending on the types of amino acid substitutions (Table 2). These mutations were predicted to affect ATP binding site motif, N-myristolylation, casein kinase II (CK2), protein kinase A (PKA) phosphorylation site and Forkhead-associated (FHA) functional sites. Mutations at the Walker A region also might led to the decrement of protein stability as predicted by I-Mutant and MuPro. Mutation at the Q-loop region (Q159H) (Figs. 2A and 2B) also predicted to have significant deleterious effect and decreased protein stability, but no effects have been identified on its molecular mechanism and structural phenotyping as predicted by MutPred and SNPeffect 4.0, respectively (Table 2).

Mild deleterious effect was predicted at the mutated zinc hook domain (Table 2; Fig. 2E). Subsequent analysis using MutPred also revealed that any mutation at zinc hook might affect several important functional sites that involved in DNA damage repair signaling response and cell cycle checkpoints such as phosphatidylinositol 3-kinase-related

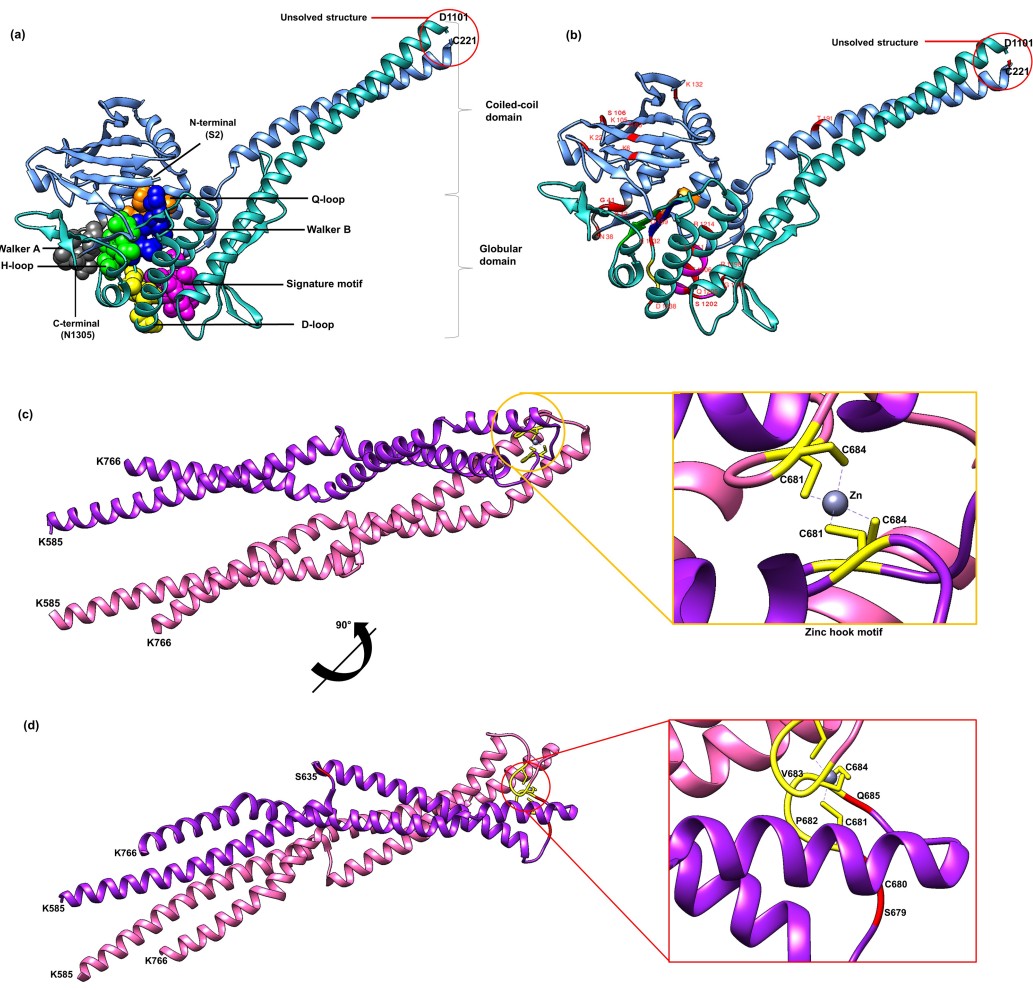

**Figure 2 3D structure of Rad50.** A 3D structure of Rad50 human modeled using fold recognition technique Phyre2 using structure from *Chaetomium thermophilum* as a template (PDB ID: 5DAC). N-terminal of 276 residues (2-278) and C-terminal of 155 residues (1153-1306) are colored as blue and green, respectively (A). All six motifs identified are marked and represented by ball and stick representation with different colors (orange for Q-loop, blue for Walker B, purple for signature motif, yellow for D-loop, gray for Walker A and green from H-loop) (A). All mutated residues identified were marked and labeled in the 3D structure (B). Zinc hook structure of 181 residues (residue 585-766) that has been determined (PDB ID: 5GOX) and its deleterious residues also marked in the structure (C and D). All figures were generated using UCSF Chimera (*Pettersen et al., 2004*).

kinases (PIKK) phosphorylation site, protein kinase C (PKC) phosphorylation site and BRCA1 C-terminus (BRCT) phosphopeptide ligands binding sites (Table 2). Moreover, deleterious mutation was predicted at the conserved cysteine residue located at the zinc hook motif (CXXC). For example, amino acid substitutions of alanine (A) and serine (S) at the cysteine residue position 681; (C681(A/S)) (Fig. 2D) may affect N-glycosylation, proline-directed phosphorylation and mitogen-activated protein kinases (MAPK) phosphorylation site, which possibly due to the affected zinc binding domain (Table 2). Another deleterious mutation, C684(R/S) was also predicted to not affect its molecular mechanism but might disrupt diarginine retention/retrieving signal, PKC and PIKK

**Table 2 In silico analysis of 42 deleterious mutations in Rad50.**

| Motif | Mutation (source) | Amino acid impact (PredictSNP) (neutral/deleterious) | | | | | | | Molecular mechanisms (MutPred2) | | | | Structural phenotyping (SNPeffect) | | Protein stability (imutant/MuPro) | |
| | | PS | MP | Ph-S | PP-1 | PP-2 | SF | SP | Affected molecular mechanisms | Pr | P-value | Affected functional sites | AG | AM | IM | MPr |
|---|---|---|---|---|---|---|---|---|---|---|---|---|---|---|---|---|
| Walker A | N38A (*De La Rosa & Nelson, 2011*) | D | D | D | D | D | D | D | Loss of catalytic site at N38 | 0.53 | 8.80E–05 | • ATP/GTP-binding site motif A (P-loop) | No | No | ↓ | ↓ |
| | | | | | | | | | Loss of relative solvent accessibility | 0.34 | 3.40E–03 | | | | | |
| | | | | | | | | | Altered ordered interface | 0.33 | 9.90E–03 | | | | | |
| | | | | | | | | | Loss of allosteric site at N38 | 0.31 | 3.00E–03 | | | | | |
| | | | | | | | | | Altered DNA binding | 0.25 | 6.70E–03 | | | | | |
| | | | | | | | | | Altered metal binding | 0.23 | 0.02 | | | | | |
| | | | | | | | | | Gain of methylation at K42 | 0.17 | 8.40E–03 | | | | | |
| | G41D (*Alani, Padmore & Kleckner, 1990*) | D | D | D | D | D | D | D | Altered metal binding | 0.40 | 1.70E–04 | • FHA phosphopeptide ligands | No | No | ↓ | ↓ |
| | | | | | | | | | Gain of allosteric site at G41 | 0.33 | 6.40E–04 | • CK2 Phosphorylation site | | | | |
| | | | | | | | | | Altered ordered interface | 0.30 | 0.02 | • N-myristoylation site | | | | |
| | | | | | | | | | Gain of relative solvent accessibility | 0.29 | 0.01 | • ATP/GTP-binding site motif A (P-loop) | | | | |
| | | | | | | | | | Gain of helix | 0.29 | 0.01 | | | | | |
| | | | | | | | | | Loss of strand | 0.28 | 9.70E–03 | | | | | |
| | | | | | | | | | Altered DNA binding | 0.28 | 4.30E–03 | | | | | |
| | | | | | | | | | Loss of catalytic site at N38 | 0.27 | 3.70E–03 | | | | | |
| | | | | | | | | | Loss of methylation at K42 | 0.17 | 0.01 | | | | | |
| | K42R (*Chen et al., 2005; Koroleva et al., 2007*) | D | D | D | D | D | D | D | Loss of relative solvent accessibility | 0.30 | 9.60E–03 | • FHA phosphopeptide ligands | No | No | ↓ | ↓ |
| | | | | | | | | | Altered DNA binding | 0.28 | 3.20E–03 | • PKA phosphorylation site | | | | |
| | | | | | | | | | Loss of allosteric site at T44 | 0.28 | 7.30E–03 | • CK2 phosphorylation site | | | | |
| | | | | | | | | | Loss of catalytic site at N38 | 0.27 | 3.40E–03 | • N-myristoylation site | | | | |
| | | | | | | | | | Altered metal binding | 0.25 | 0.01 | • ATP/GTP-binding site motif A (P-loop) | | | | |
| | | | | | | | | | Loss of methylation at K42 | 0.20 | 6.90E–03 | | | | | |
| | K42M (*Koroleva et al., 2007*) | D | D | D | D | D | D | D | Altered DNA binding | 0.37 | 7.10E–04 | • FHA phosphopeptide ligands | No | No | ↑ | ↑ |
| | | | | | | | | | Loss of allosteric site at K42 | 0.36 | 1.10E–03 | • CK2 phosphorylation site | | | | |
| | | | | | | | | | Loss of relative solvent accessibility | 0.34 | 3.10E–03 | • N-myristoylation site | | | | |

(Continued)

| Motif | Mutation (source) | Amino acid impact (PredictSNP) (neutral/deleterious) | | | | | | | Molecular mechanisms (MutPred2) | | | Affected functional sites | Structural phenotyping (SNPeffect) | | Protein stability (imutant/MuPro) | |
|---|---|---|---|---|---|---|---|---|---|---|---|---|---|---|---|---|
| | | PS | MP | Ph-S | PP-1 | PP-2 | SF | SP | Affected molecular mechanisms | Pr | P-value | | AG | AM | IM | MPr |
| | | | | | | | | | Altered ordered interface | 0.32 | 0.01 | • ATP/GTP-binding site motif A (P-loop) | | | | → |
| | | | | | | | | | Gain of catalytic site at T43 | 0.29 | 1.60E–03 | | | | | |
| | | | | | | | | | Altered metal binding | 0.28 | 5.80E–03 | | | | | |
| | | | | | | | | | Loss of methylation at K42 | 0.20 | 6.80E–03 | | | | | |
| | K42E (*Alani, Padmore & Kleckner, 1990; Chen et al., 2005*) | D | D | D | D | D | D | D | Altered metal binding | 0.44 | 4.50E–04 | • FHA phosphopeptide ligands | No | No | → | |
| | | | | | | | | | Gain of catalytic site at T43 | 0.33 | 6.40E–04 | • CK2 phosphorylation site | | | | |
| | | | | | | | | | Altered DNA binding | 0.33 | 1.30E–03 | • Polo-like kinase phosphorylation sit | | | | |
| | | | | | | | | | Loss of allosteric site at K42 | 0.30 | 4.30E–03 | • N-myristoylation site | | | | |
| | | | | | | | | | Altered ordered interface | 0.29 | 0.02 | • ATP/GTP-binding site motif A (P-loop) | | | | |
| | | | | | | | | | Loss of relative solvent accessibility | 0.29 | 0.01 | | | | | |
| | | | | | | | | | Gain of strand | 0.27 | 0.03 | | | | | |
| | | | | | | | | | Loss of methylation at K42 | 0.20 | 6.80E–03 | | | | | |
| | K42A (*Chen et al., 2005*) | D | D | D | D | D | D | D | Loss of allosteric site at K42 | 0.53 | 5.40E–05 | • FHA phosphopeptide ligands | No | No | → | → |
| | | | | | | | | | Loss of relative solvent accessibility | 0.37 | 1.60E–03 | • CK2 phosphorylation site | | | | |
| | | | | | | | | | Altered DNA binding | 0.37 | 6.00E–04 | • N-myristoylation site | | | | |
| | | | | | | | | | Altered ordered interface | 0.36 | 4.20E–03 | • ATP/GTP-binding site motif A (P-loop) | | | | |
| | | | | | | | | | Gain of catalytic site at T43 | 0.32 | 8.70E–04 | | | | | |
| | | | | | | | | | Altered metal binding | 0.31 | 0.01 | | | | | |
| | | | | | | | | | Loss of methylation at K42 | 0.20 | 6.80E–03 | | | | | |
| Q-loop | Q159H (*Moncalian et al., 2004*) | D | D | D | D | D | D | D | No effect | – | – | None | No | No | → | → |
| Zinc hook | S635G (*Gatei et al., 2011*) | N | N | N | D | N | N | N | No effect | – | – | None | No | No | → | → |
| | S679R (*Roset et al., 2014; Hohl et al., 2015*) | N | N | N | N | N | D | N | No effect | – | – | None | No | No | → | → |
| | C680N (*He et al., 2012*) | N | N | N | D | N | D | D | Loss of N-linked glycosylation at N677 | 0.02 | 0.04 | • N-glycosylation site | No | No | → | → |

| Motif | Mutation (source) | Amino acid impact (PredictSNP) (neutral/deleterious) | | | | | | | Molecular mechanisms (MutPred2) | | | | Structural phenotyping (SNPeffect) | | Protein stability (imutant/MuPro) | |
|---|---|---|---|---|---|---|---|---|---|---|---|---|---|---|---|---|
| | | PS | MP | Ph-S | PP-1 | PP-2 | SF | SP | Affected molecular mechanisms | Pr | P-value | Affected functional sites | AG | AM | IM | MPr |
| | C681A (*He et al., 2012*) | D | D | D | D | D | D | D | Gain of helix | 0.30 | 8.00E−03 | • N-glycosylation site | No | No | ↓ | ↓ |
| | | | | | | | | | Gain of N-linked glycosylation at N677 | 0.02 | 0.03 | | | | | |
| | C681S (*Barfoot et al., 2015*) | D | D | D | D | D | D | N | Gain of N-linked glycosylation at N677 | 0.02 | 0.04 | • N-glycosylation site | No | No | ↑ | ↓ |
| | | | | | | | | | | | | • Proline-directed phosphorylation | | | | |
| | | | | | | | | | | | | • MAPK phosphorylation site | | | | |
| | P682E (*Roset et al., 2014; Hohl et al., 2015*) | D | D | N | D | D | D | D | Gain of helix | 0.32 | 3.10E−03 | • CK2 phosphorylation site | No | No | ↓ | ↓ |
| | | | | | | | | | Altered coiled coil | 0.14 | 0.03 | | | | | |
| | | | | | | | | | Loss of N-linked glycosylation at N677 | 0.02 | 0.04 | | | | | |
| | P682A (*He et al., 2012*) | N | N | N | N | D | D | N | No effect | – | – | None | No | No | ↓ | ↓ |
| | V683I (*He et al., 2012*) | N | N | N | N | N | D | N | No effect | – | – | None | No | No | ← | ← |
| | V683R (*Roset et al., 2014; Hohl et al., 2015*) | N | N | N | N | N | D | D | Gain of helix | 0.29 | 0.01 | None | No | No | ↓ | ↓ |
| | C684R (*He et al., 2012*) | D | D | D | D | D | D | D | Gain of helix | 0.30 | 9.50E−03 | • Diarginine retention/retrieving signal | No | No | ↓ | ↓ |
| | C684S (*Barfoot et al., 2015*) | D | D | D | D | D | D | D | No effect | – | – | • PIKK phosphorylation site | No | No | ↓ | ↓ |
| | | | | | | | | | | | | • PKC phosphorylation site | | | | |
| | Q685S (*He et al., 2012*) | N | N | N | N | N | D | N | Altered coiled coil | 0.53 | 6.50E−03 | • BRCT phosphopeptide ligands | No | No | ↓ | ↓ |
| | | | | | | | | | | | | • USP7 binding motif | | | | |
| Signature motif | R1198E (*Rojowska et al., 2014*) | D | D | D | D | D | D | D | Gain of catalytic site at R1200 | 0.25 | 4.80E−03 | • Diarginine retention/retrieving signal | No | No | ↓ | ↓ |
| | | | | | | | | | Gain of allosteric site at R1200 | 0.21 | 0.03 | | | | | |
| | | | | | | | | | Altered metal binding | 0.14 | 0.03 | | | | | |
| | | | | | | | | | Altered transmembrane protein | 0.10 | 0.04 | | | | | |
| | G1199E (*Rojowska et al., 2014*) | D | D | D | D | D | D | D | Loss of allosteric site at R1200 | 0.23 | 0.03 | • Diarginine retention/retrieving signal | No | No | ↓ | ↓ |
| | | | | | | | | | Loss of catalytic site at R1200 | 0.20 | 0.01 | • PKA phosphorylation site | | | | |
| | | | | | | | | | Altered transmembrane protein | 0.11 | 0.04 | | | | | |

(Continued)

| Motif | Mutation (source) | Amino acid impact (PredictSNP) (neutral/deleterious) | | | | | | | Molecular mechanisms (MutPred2) | | | Affected functional sites | Structural phenotyping (SNPeffect) | | Protein stability (imutant/MuPro) | |
|---|---|---|---|---|---|---|---|---|---|---|---|---|---|---|---|---|
| | | PS | MP | Ph-S | PP-1 | PP-2 | SF | SP | Affected molecular mechanisms | Pr | P-value | | AG | AM | IM | MPr |
| | S1202A (*Herdendorf & Nelson, 2011*) | D | D | D | D | D | D | D | Loss of allosteric site at R1200 | 0.23 | 0.03 | • PKA phosphorylation site | No | No | → | → |
| | | | | | | | | | Loss of catalytic site at R1200 | 0.20 | 0.01 | • Glycosaminoglycan attachment site | | | | |
| | S1202R (*Koroleva et al., 2007; Moncalian et al., 2004; Herdendorf & Nelson, 2011; Bhaskara et al., 2007*) | D | D | D | D | D | D | D | Gain of ADP-ribosylation at S1202 | 0.25 | 8.40E−03 | • PKA phosphorylation site | No | No | ↑ | → |
| | | | | | | | | | Loss of allosteric site at R1200 | 0.23 | 0.03 | • Glycosaminoglycan attachment site | | | | |
| | | | | | | | | | Loss of catalytic site at R1200 | 0.21 | 1.00E−02 | | | | | |
| | S1202M (*Herdendorf & Nelson, 2011*) | D | D | D | D | D | D | D | Loss of allosteric site at R1200 | 0.23 | 0.02 | • PKA phosphorylation site | No | No | ↑ | ↑ |
| | | | | | | | | | Gain of catalytic site at R1200 | 0.21 | 9.70E−03 | • Glycosaminoglycan attachment site | | | | |
| | Q1205E (*Herdendorf & Nelson, 2011*) | D | D | D | N | D | D | D | Gain of allosteric site at R1200 | 0.23 | 0.02 | • PKA phosphorylation site | ↑ | → | → | → |
| | | | | | | | | | Loss of catalytic site at R1200 | 0.20 | 0.01 | • CK2 phosphorylation site | | | | |
| | K1206M (*Herdendorf & Nelson, 2011*) | D | D | D | D | D | D | D | Gain of catalytic site at S1202 | 0.09 | 0.04 | None | ↑ | → | ↑ | ↑ |
| | K1206A (*Williams et al., 2011*) | D | D | D | D | D | D | D | Loss of catalytic site at K1206 | 0.09 | 0.05 | None | ↑ | → | → | ↑ |
| | K1206E (*Williams et al., 2011*) | D | D | D | D | D | D | D | Gain of catalytic site at K1206 | 0.11 | 0.03 | • TRAF2 binding site | ↑ | → | → | ↑ |
| | | | | | | | | | | | | • NES nuclear export signal | | | | |
| | L1211W (*Deshpande et al., 2014*) | D | D | D | D | D | D | D | Loss of catalytic site at K1206 | 0.08 | 0.05 | • SUMO interaction site | ↑ | ↑ | → | → |
| | R1214A (*Williams et al., 2011*) | D | D | D | D | D | D | D | Loss of allosteric site at R1214 | 0.22 | 0.03 | • ATP-binding cassette, ABC transporter-type, signature and profile | ↑ | → | → | ↑ |
| | R1214E (*Deshpande et al., 2014; Williams et al., 2011*) | D | D | D | D | D | D | D | Loss of allosteric site at R1214 | 0.20 | 0.04 | • SUMO interaction site | ↑ | No | → | → |
| | | | | | | | | | | | | • ATP-binding cassette, ABC transporter-type, signature and profile | | | | |

| Motif | Mutation (source) | Amino acid impact (PredictSNP) (neutral/deleterious) | | | | | | | Molecular mechanisms (MutPred2) | | | Affected functional sites | Structural phenotyping (SNPeffect) | | Protein stability (imutant/MuPro) | |
|---|---|---|---|---|---|---|---|---|---|---|---|---|---|---|---|---|
| | | PS | MP | Ph-S | PP-1 | PP-2 | SF | SP | Affected molecular mechanisms | Pr | P-value | | AG | AM | IM | MPr |
| Walker B | E1232Q (*Rojowska et al., 2014*) | D | D | D | D | D | D | D | Altered metal binding | 0.48 | 4.30E−03 | • FHA phosphopeptide ligands | No | No | ↓ | ↓ |
| | | | | | | | | | Loss of catalytic site at E1232 | 0.34 | 9.80E−04 | • SUMO interaction site | | | | |
| | | | | | | | | | Loss of allosteric site at P1233 | 0.24 | 0.02 | | | | | |
| | | | | | | | | | Altered transmembrane protein | 0.12 | 0.03 | | | | | |
| D-loop | D1238N (*De La Rosa & Nelson, 2011*) | D | D | D | D | D | D | D | Altered ordered interface | 0.30 | 4.30E−03 | • FHA phosphopeptide ligands | No | No | ↓ | ↓ |
| | | | | | | | | | Altered metal binding | 0.31 | 2.80E−03 | • Casein kinase II phosphorylation site | | | | |
| | | | | | | | | | Gain of relative solvent accessibility | 0.27 | 0.02 | | | | | |
| | | | | | | | | | Gain of allosteric site at P1233 | 0.25 | 0.01 | | | | | |
| | | | | | | | | | Loss of catalytic site at T1234 | 0.17 | 0.02 | | | | | |
| | | | | | | | | | Altered transmembrane protein | 0.12 | 0.03 | | | | | |
| | | | | | | | | | Altered coiled coil | 0.08 | 0.05 | | | | | |
| | D1238A (*De La Rosa & Nelson, 2011*) | D | D | D | D | D | D | D | Altered metal binding | 0.41 | 3.40E−04 | • FHA phosphopeptide ligands | No | No | ↓ | ↓ |
| | | | | | | | | | Altered ordered interface | 0.40 | 1.40E−03 | • Casein kinase II phosphorylation site | | | | |
| | | | | | | | | | Loss of allosteric site at P1233 | 0.26 | 0.01 | | | | | |
| | | | | | | | | | Loss of catalytic site at T1234 | 0.18 | 0.02 | | | | | |
| | | | | | | | | | Altered transmembrane protein | 0.12 | 0.02 | | | | | |
| ATPase domain/ coiled-coil | K6E (*Alani, Padmore & Kleckner, 1990; Bender et al., 2002*) | D | D | D | D | D | N | D | Loss of strand | 0.27 | 0.03 | None | No | No | ↓ | ↓ |
| | | | | | | | | | Altered DNA binding | 0.16 | 0.04 | | | | | |
| | | | | | | | | | Gain of N-terminal acetylation at M1 | 0.03 | 4.10E−03 | | | | | |
| | K22M (*Alani, Padmore & Kleckner, 1990; Bender et al., 2002*) | N | N | N | N | N | D | N | No effect | – | – | None | ↑ | No | ↓ | ↑ |
| | R83I (*Alani, Padmore & Kleckner, 1990; Bender et al., 2002*) | N | D | N | N | N | D | N | Altered ordered interface | 0.29 | 0.03 | • PP1-docking motif RVXF | ↑ | ↑ | ↓ | ↑ |
| | | | | | | | | | Altered DNA binding | 0.22 | 0.02 | | | | | |
| | | | | | | | | | Altered coiled coil | 0.10 | 0.04 | | | | | |

(Continued)

| Motif | Mutation (source) | Amino acid impact (PredictSNP) (neutral/deleterious) | | | | | | | Molecular mechanisms (MutPred2) | | | Affected functional sites | Structural phenotyping (SNPeffect) | | Protein stability (imutant/MuPro) | |
|---|---|---|---|---|---|---|---|---|---|---|---|---|---|---|---|---|
| | | PS | MP | Ph-S | PP-1 | PP-2 | SF | SP | Affected molecular mechanisms | Pr | P-value | | AG | AM | IM | MPr |
| | K132E (*Rojowska et al., 2014*) | D | D | N | D | D | D | D | Loss of helix | 0.28 | 0.02 | • CK1 phosphorylation site | No | No | ↓ | ↓ |
| | | | | | | | | | Altered transmembrane protein | 0.27 | 7.30E−04 | • Protein kinase C phosphorylation site | | | | |
| | | | | | | | | | Gain of strand | 0.27 | 0.01 | | | | | |
| | T191E (*Rojowska et al., 2014*) | N | D | N | N | N | N | N | Altered coiled coil | 0.28 | 0.01 | • TRAF2 binding site | No | No | ↓ | ↓ |
| | | | | | | | | | Loss of acetylation at K187 | 0.28 | 6.20E−03 | • NEK2 phosphorylation site | | | | |
| | | | | | | | | | | | | • PKC phosphorylation site | | | | |
| | C221E (*Rojowska et al., 2014*) | N | N | N | N | N | N | N | No effect | – | – | None | No | No | ↓ | ↓ |
| | K105E (*Rojowska et al., 2014*) | D | N | D | D | D | D | D | No effect | – | – | None | No | No | ↓ | ↓ |
| | S106E (*Rojowska et al., 2014*) | N | N | N | N | N | N | N | No effect | – | – | None | No | No | ↑ | ↓ |

**Note:**
Different tools were used to analyze all mutations as abbreviated in the table. PS, PredictSNP; MP, MAPP; PhS, PhD-SNP; PP1, Poly-Phen1; PP2, Poly-Phen2; SF, SIFT; SN, SNAP; IM, I-Mutant; MPr, MuPro; Pr, probability; AG, protein aggregation; AM, Amyloid aggregation. Please refer to "Materials and Methods" for detailed descriptions of these tools. Note that all mutations listed above are based on the equivalent mutations in human.

phosphorylation site (Table 2). Whilst P682E (Fig. 2D) mutation may lead to gain of helix, altered coiled coil domain, loss of N-linked glycosylation and CK2 phosphorylation site (Table 2).

Rad50 signature motif (Fig. 2A) is a critical site which could lead to deleterious effects if mutated as suggested by PredictSNP analysis (Table 2). All mutations in this motif (S1202A/R/M, Q1205E and K1206M/A/E) or located near this motif (G1198E, L1211W and R1214A/E) (Figs. 1B and 2B) were predicted to affect the protein allosteric and catalytic sites (Table 2), except for R1198E. Mutations at residue S1202A/R/M (Figs 1B and 2B) might affect PKA phosphorylation sites and glycosaminoglycan attachment site (Table 2). Furthermore, R1214A (Figs. 1B and 2B) mutation might affect ATP-binding cassette, ABC transporter-type, signature and profile functional sites (Table 2). We have also predicted several mutations in Rad50 signature motif such as Q1205E, L1211W and R1214A that contributed to the total defect in the structural phenotyping such as the increment in protein and amyloid aggregation and the decrement of protein stability (Table 2).

We have also predicted K6E, K132E and K105E mutations occurred at the coil-coiled domain or ATPase domain to be deleterious (Figs. 1B and 2B; Table 2). Specifically, the mutations at K6E and K132E might lead to loss of strand or loss of helix, respectively. Additionally mutation at K132E also predicted to affect casein kinase 1 (CK1) and PKC phosphorylation sites (Table 2). Even though K22M and R83I (Fig. 2B) were predicted to be neutral in PredictSNP analysis, both of these mutations have also been predicted to increase protein aggregation tendency (Table 2). The mutation at R83I might contributed to the alteration of coiled coil structure domain, DNA binding and ordered interface, that might affect the functional site such as protein–protein interactions (PPI)-docking motif (Table 2). Another neutral mutation predicted were T191E, C221E and S106E (Figs. 1B and 2B), where T191E mutation might be responsible in altering the coiled coil domain and may affect tumor necrosis factor receptor-associated factor (TRAF), serine/threonine-protein kinase (NEK2) and PKC phosphorylation site (Table 2). On the other hand, C221E and S106E (Figs. 1B and 2B) were predicted to not affect any molecular mechanism or protein aggregation (Table 2).

## DISCUSSION

Rad50 is a member of the structural maintenance of chromosomes (SMC) family of proteins that participates in chromosome structural changes (*Kinoshita et al., 2009*). The globular ABC ATPase head domain is formed by the N- and C-termini (Fig. 2A) (*Hohl et al., 2011*). The coiled-coil apex of Rad50 contains a conserved cysteine amino acid motif across the organisms, which is called the zinc hook (*Kinoshita et al., 2009*). When DNA double strand break occurs, Rad50 complex binds to the DNA early in the repair process to recognize such breaks and grips them in close juxtaposition (*Paull & Gellert, 1998*; *De Jager et al., 2001*). This protein also activates ATM kinase that is crucial for DNA damage signaling (*Uziel et al., 2003*).

Rad50 globular head domain contains conserved domains and motifs (Figs. 1A and 2B) such as P-loop NTPase domains and six motifs which are Walker A and B motifs, Rad50

signature motif, D-loop, H-loop, and Q-loop motif (Figs. 1A and 2B). P-loop NTPase domains are belong to ABC protein superfamily. The ABC protein superfamily has been identified in diverse organisms and is also known to be one of the most conserved protein superfamilies (*Jones, O'Mara & George, 2009*). ABC proteins consist of six conserved motifs (Figs. 1A and 2A) which make up the nucleotide binding domain in Rad50 (*Symington, 2002*). The nucleotide binding domain of ABC protein is known to play an important role in binding and hydrolyzing ATP at its dimeric interface (*Davidson et al., 2008*). Rad50 also has a special conserved Cys-X-X-Cys zinc hook motif at the center of coiled-coil domain (Fig. 2C). This motif is in a hook-shaped structure which dimerizes a second hook via cysteine-mediated zinc ion coordination (Fig. 2C) (*Hopfner et al., 2002*). This zinc dependent dimerization event allows the formation of MRN complex which has suitable lengths and conformational arrangements to link sister chromatids in HR and DNA ends in NHEJ (*Hopfner et al., 2002*).

## Consistency between bioinformatics prediction and experimental evidence

PredictSNP was used in this study to provide a more accurate prediction of disease-related mutations as it combines six best performing prediction tools for a consensus classifier (*Bendl et al., 2014*). Evidently, this in silico analysis was consistent with the results from the previous experimental studies where mutations at the Walker A, D-loop, signature motif, Q-loop and Walker B have shown damaging effects (Tables 1 and 2 ; Fig. 2A).

G41D and K40E (Figs. 1B and 2B) mutations at the Walker A motif (Fig. 2A) and C681A and C684R (Figs. 1B and 2D) mutations at the cysteine residue (CXXC) in the zinc hook motif (Fig. 2C) conferred an identical phenotype with the Rad50 null mutation characterized by total defect in the formation of viable spore in *S. cerevisiae* experiment (Table 1) (*Alani, Padmore & Kleckner, 1990*; *He et al., 2012*). This analysis also identified that mutations at Q-loop (Q159H) and D-loop (D1238N and D1238A) (Figs. 1B, 2A and 2B) were also predicted deleterious (Table 2) and were experimentally shown to interrupt all ATP-dependent activities of the complex in different organisms such as *P. furiosus* and T bacteriophage respectively (Table 1) (*Moncalian et al., 2004*; *De La Rosa & Nelson, 2011*). Furthermore, a E1232Q (Figs. 1B and 2B) mutation at the Walker B motif (Fig. 2A) was also predicted to be deleterious (Table 2). Similarly the mutation of Walker B at residue E798Q in *Thermotoga maritima* showed low ability to respond to DNA damage (Table 1) (*Rojowska et al., 2014*). This suggests that this motif is important for a molecular repair process, specifically during DNA binding process, which if mutated will affect the viability of an organism. Our analysis using PredictSNP has identified three mutations, which were N28A (Figs. 1B and 2B) (*De La Rosa & Nelson, 2011*), D1238H (Figs. 1B and 2B) (*De La Rosa & Nelson, 2011*) and S1202R (Figs. 1B and 2B) (*Kerem et al., 1989*; *Moncalian et al., 2004*) located at the Walker A, D-loop and Rad50 signature motif, respectively (Fig. 2A) (*Kerem et al., 1989*; *Moncalian et al., 2004*; *De La Rosa & Nelson, 2011*).

Mutations at the Walker A domain and Rad50 signature motif (Fig. 2A) may also affect important functional sites such as ATP binding site (Table 2). For example, K42R/M/E/A

mutation at the Walker A (Figs. 1B and 2B) in *S. cerevisiae* and *D. radiodurans* has been identified experimentally to cause defective in ATPase (Table 1) (*Chen et al., 2005*; *Koroleva et al., 2007*) and S793R mutation in *Pyrococcus furiosus* showed the inhibition of ATP binding and disrupted communication between ATP loops (Table 1) (*Moncalian et al., 2004*). This mutation further distorted the surface of the C-terminal domain and thus altered the interaction between Rad50 monomers to prevent dimerization (Table 1) (*Moncalian et al., 2004*). We have also identified mutations at several motifs such as Walker A (G41D, K42M/R/E/A) and Walker B (E1232Q) (Figs. 1B, 2A and 2B) that might affect the binding of FHA phosphopeptide ligands that plays a critical role in DNA damage repair mechanism and cell cycle (Table 2). Many FHA domain–containing proteins localized to the nucleus showed to play a critical role in establishing or maintaining DNA repair, cell cycle checkpoints or transcriptional regulation (*Durocher et al., 2000*). When mutated, diseases such as Nijmegen breakage syndrome (NBS) and the hereditary cancer syndrome variant Li-Fraumeni (CHK2) will be developed (*Matsuura et al., 1998*; *Varon et al., 1998*; *Carney et al., 1998*; *Featherstone & Jackson, 1998*; *Bell et al., 1999*) suggesting the importance of these conserved residues within Rad50 for DNA repair and maintenance.

Mutations at or near the Rad50 signature motif (Figs. 1B and 2A) were also known to be damaging (Table 2), particularly the S1202R (Figs. 1B and 2B) mutation which has been studied the most due to its numerous biological defects in vivo. The same residue mutations of the Rad50 signature motif in yeast (S12025R) and human (S1202R) also generated complexes that were significantly diminished in adenylate kinase (AK) activity that was important for DNA tethering (*Bhaskara et al., 2007*). Previously, AK deficiency was found to be associated with anemia and several cases of mental retardation and psychomotor impairment (*Abrusci et al., 2007*), which may explain why disruption of the MRN complex also causes this phenotype on patients (*Waltes et al., 2009*). In addition, such deleterious mutation also contributed to inviable spores and significant telomere shortening in *S. cerevisiae* (*Bhaskara et al., 2007*). Defects in telomere length in human have been known to cause the pathology of several age-related diseases and premature aging syndrome, as well as cancer and other human diseases such as Hoyeraal-Hreidarsson syndrome, Coats plus syndrome, pulmonary fibrosis, dyskeratosis congenita, liver fibrosis and aplastic anemia (*Blasco, 2005*).

Additionally, most mutations such as G1199E, S1202A/R/M, and Q1205E (Figs. 1B and 2B) at the Rad50 signature motif (Fig. 2A) were identified to affect PKA phosphorylation site (Table 2) suggesting that this site is dependent upon the function of Rad50 signature motif. Phosphorylation is one of the most ubiquitous and important post translational modifications of proteins, and implicated in almost all kinds of cellular processes and pathways (*Ptacek & Snyder, 2006*). In neurons, enhanced PKA signaling promotes neuronal development, enhances synaptic plasticity, and elevates dopamine synthesis (*Dagda & Das Banerjee, 2015*). However a deterioration in PKA signaling has contributed to the etiology of several neurodegenerative diseases, such as Alzheimer and Parkinson (*Dagda & Das Banerjee, 2015*). We hypothesized that the defective PKA functional sites may also lead to Nijmegen breakage syndrome associated with
neurological phenotype in Rad50 mutations (*Waltes et al., 2009*), however this potential phosphorylation sites remain to be validated.

C681A and C684R mutations (Figs. 1B and 2D) at the zinc hook motif (Fig. 2C) were identified deleterious from our analysis (Table 2) and these mutations were known to lead severe defects in various DNA damage response (DDR) such as ataxia-telangiectasia mutated (ATM) protein activation, homologous recombinant, irradiation sensitivity and ataxia telangiectasia and Rad3 related (ATR) protein activation (*He et al., 2012*). These findings were consistent with our bioinformatics analysis where C684S deleterious mutation at zinc hook (Figs. 1B and 2D) might affect a protein kinase called ataxia telangiectasia mutated (ATM) that belongs to the phosphatidylinositol 3-kinase-related kinase (PIKK) family (Table 2). The ATM protein was known to cause devastating ataxia-telangiectasia syndrome which is characterized by progressive neurological disorder, impaired organ maturation and immunodeficiency (*Shiloh & Ziv, 2013*). Rad50 phosphorylated ATM at S635 site (Figs. 1B and 2D) of which the mutation on this site showed its importance for cell cycle control signaling and DNA repair mechanism (*Gatei et al., 2011*).

P682E mutation at the zinc hook motif (Figs. 1B and 2D) was shown to be deleterious (Table 2), where previous study has reported that the double mutation P682E and S679R at the zinc hook motif have reduced zinc affinity and dimerization efficiency leading to mice lethality (*Roset et al., 2014*). In addition, crossbreeding P682E and S679R mutant mice with wildtype mice produce offsprings with hydrocephalus (accumulation of cerebrospinal fluid within the brain), defects in hematopoietic stem cells and gametogenic cells. This suggests that the hook motif has strong influence on the MRN complex associated with DDR signaling, tissue homeostasis and tumorigenesis, as well as fertility of the organism (*Roset et al., 2014*). This is consistent with the mutations in the yeast hook domain that has increased chromosomal fragmentation (*Cahill & Carney, 2007*), suggesting its presence is required for the binding or tethering of chromosomal ends.

## Limitations of in silico prediction

Several mutations were functionally predicted to be neutral, in contrast with the previous experimental findings. For example, a few mutations that is, S635G (*H. sapiens*), S679R, C680N, P682A, V683I (*S. cerevisiae*), V683R (*M. musculus*) and Q685S (*S. cerevisiae*) (Figs. 1B and 2D) located at the zinc hook domain (Table 2) and mutations on K22M (*S. cerevisiae* and *M. musculus*), R83I (*S. cerevisiae* and *M. musculus*), T191E, C221E and S106E (*T. maritima*) in the ATPase domain (Figs. 1B and 2B; Table 2) were experimentally validated to be deleterious; some causing embryonic lethality, growth defect, cancer predisposition, as well as hematopoietic and spermatogenic depletion in vivo (*Bender et al., 2002*). A few previous studies have also shown discrepancies between computer prediction and experimental data. For example, an extensive in silico analysis using PolyPhen2 and MutPred tools of the ATP-binding cassette transporter ABCA1, an important target in anti-atherosclerosis treatment predicted that several nsSNPs can be neutral, contradicting with previous experimental data findings (*Marín-Martín et al., 2014*). Furthermore, another in silico analysis performed using PolyPhen and SIFT on proteins related to

several hereditary diseases such as glucose-6-phosphate dehydrogenase deficiency (G6PD), the receptor 1 for tumor necrosis factor-(TNFRSF1A), and familial mediterranean fever (MEFV) has concluded that some nsSNPs impact may also not be predicted deleterious to correspond to previous phenotypic effect (*Tchernitchko, Goossens & Wajcman, 2004*). Moreover, in silico identification of PmrAB virulence targets in *Salmonella typhimurium* also demonstrated false positive prediction when validated experimentally (*Marchal et al., 2004*) suggesting that more work has to be done to develop a more accurate bioinformatics prediction platforms in the future. In contrast, various SNP prediction software have predicted that these mutations were not damaging (Table 2).

Such discrepancy between the computational prediction and experimental results may be due to several limitations in the bioinformatic tools used in our analysis. Several web-based prediction tools may supply conflicting results (*Wan et al., 2008*) and even with an integrated predictor, PredictSNP (*Bendl et al., 2014*), it is also limited by the differences in algorithms, principles, training datasets and information used. For example, MAPP, PANTHER and SIFT in the PredictSNP used alignment scores for functional prediction whereas SNAP, PoplyPhen-1 and PolyPhen-2 used neural network, support vector machine and Naïve Bayes algorithm, respectively (*Bendl et al., 2014*). Interestingly, we identified that the software predicts most accurately (in agreement with experimental results) for the motifs or sites located at the highly conserved position (Fig. 1B). Conversely, most residues that were predicted to be neutral are located at non-conserved positions in the Rad50 protein (Fig. S3; Table S7), suggesting that these prediction software may have only been trained and preferentially biased towards conserved regions (*Gardner et al., 2017*). This suggests that computer prediction should also consider and take into account the effect of non-conserved regions outside the motifs/domains too for future improvement in their algorithms. Furthermore, any subsequent prediction studies should also be aware of this limitation (whether located in conserved or non-conserved regions) to carefully deduce the function of their protein mutation of interest.

Nonetheless, we cannot rule out the possibility that these mutations derived from other organisms may not be readily affected or transferred to other organisms including human. This is because certain organisms may possess gene compensation to compensate or mask the effect of such mutations and that the different proteins from different organisms may not have perfectly superimposable function. Hence, future experiments should focus on their validation especially in human cell line studies to better understand the roles of these mutated residues in Rad50 function.

## CONCLUSIONS

This study compiled all mutations to date in Rad50 proteins from various organisms and predicts their effects using various software tools such as PredictSNP, MutPred, SNPeffect, I-Mutant and MUpro. Most predictions for SNPs occurring within conserved regions are in agreement with their corresponding in vivo or in vitro experimental results. However, SNPs located at non-conserved regions are less likely to be accurately

predicted, and as such algorithms for these software should be improved in future studies. Altogether this study has provided means to prioritized mutations particularly in Rad50 protein that have biologically meaningful function for DNA double-stranded maintenance.

## ACKNOWLEDGEMENTS

The authors would like to acknowledge and thank INBIOSIS and Makmal Genomik 1 research teams from Universiti Kebangsaan Malaysia for their help with laboratory and technical analyses.

### Funding

This work was funded by the Ministry of Science, Technology and Innovation, Malaysia's Science Fund (Grant number: 02-01-02-SF1279). The funders had no role in study design, data collection and analysis, decision to publish, or preparation of the manuscript.

### Grant Disclosures

The following grant information was disclosed by the authors:
Ministry of Science, Technology and Innovation, Malaysia's Science Fund: 02-01-02-SF1279.

### Competing Interests

The authors declare that they have no competing interests.

### Author Contributions

- Juwairiah Remali conceived and designed the experiments, performed the experiments, analyzed the data, prepared figures and/or tables, authored or reviewed drafts of the paper, and approved the final draft.
- Wan Mohd Aizat conceived and designed the experiments, authored or reviewed drafts of the paper, and approved the final draft.
- Chyan Leong Ng conceived and designed the experiments, authored or reviewed drafts of the paper, and approved the final draft.
- Yi Chieh Lim conceived and designed the experiments, authored or reviewed drafts of the paper, and approved the final draft.
- Zeti-Azura Mohamed-Hussein conceived and designed the experiments, authored or reviewed drafts of the paper, and approved the final draft.
- Shazrul Fazry conceived and designed the experiments, authored or reviewed drafts of the paper, and approved the final draft.

### Data Availability

The raw data is available in the Supplemental Files.

## Supplemental Information

Supplemental information for this article can be found online at http://dx.doi.org/10.7717/peerj.9197#supplemental-information.

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
