# Peer review of "In silico analysis on the functional and structural impact of Rad50 mutations involved in DNA strand break repair"

_PeerJ, doi:10.7717/peerj.9197_

## Round 0.1 · original submission · Major Revisions

I would be glad to consider a substantial major revision of the manuscript addressing the important points raised by reviewer #1.

Reviewer 1 ·

Basic reporting

In this paper the authors collected known mutations for the Rad50 gene from either literature or a SNP database and annotated them using bioinformatics tools and databases, gathering information on how likely they are to be pathogenic, on the potential mechanism(s) of pathogenicity and on predicted changes in stability. They also investigated the structure by generating a partial model of the structure of Rad50 and annotated different predicted structural domains.

The paper is mostly clearly written, however the language and flow could be improved in some areas, especially regarding the Methods which are a bit short. Lines 223-234 are not detailed enough, (for instance 226-227 - what does consolidating similar mutations mean?). I especially find it difficult to understand how these mutations from different organisms are put together to gain a more general understanding (see below).

The authors generate a structural model of part of the Rad50 protein, however they don’t really use it to discuss the results and so it’s left hanging. I’d recommend integrating it better in the discussion.

The results section is short and mostly a summary of Table 2, but I feel their work could be better contextualised (see below).

Regarding table 2, the authors lists a number of functional sites possibly affected by the mutations ("Affected functional sites"), however it's not clear to me where this information comes from.

No raw data are provided in machine-readable format, even though most data (maybe except the full MSA) are present in form of word documents.

Experimental design

The aim of the study is to “identify the functional and structural effects of amino acid mutations in Rad50 gathered from exhaustive literature review and SNP database search”. While the matter is clear, it feels more like a review article than an original research article; the authors gather known SNPs/mutations for Rad50 and annotate them using some bioinformatics tool. While there is some content in their work, it feels like the research question is only superficially explored as no other validation of their findings is performed.

The researchers collect mutations of Rad50 in human and ortholog proteins, put them all together in the same pool and carry on with their analysis. I feel like a strong assumption is being made here: all mutations are taken together, without considering that they actually do come from different organisms and they are then discussed as a bulk, as if they were from a single organism or of exactly the same protein. This makes discussing the “structural and functional effects of aminoacid mutations” challenging, since it ignores the fact that a mutation might behave differently in different organisms because the protein and the more overall cellular and physiological environment are different and this should be taken into account. For instance, a mutation that is damaging in yeast Rad50 could be compensated by another substitution in human. The authors should state more clearly if they are considering a reference organism and contextualise their findings in this direction, or can make a better point of the fact that the mutations are affecting all the proteins in the family in the same way.

The authors discuss only 43 of the identified mutations since are those that have been seen to have a more important effect in the phenotype. However - again, the question remains if they are damaging in the phenotypes of all the organisms they have considered or only one/some, and/or if some of the other discarded mutations could actually be pathogenic in other organisms.

Validity of the findings

The researchers use a consensus prediction tool to assess pathogenicity. From reading the paper, I’m not sure whether their prediction are performed applying non-human mutations to the human sequence (the legend of table 2 is not very clear on this matter) or if the original organism’s wild-type protein sequence was used as a reference. If so, it should be noted that many of the tools were explicitly trained and validated on data from humans and are designed to work on human proteins, therefore it raises the question of how transferable they are to sequences from other species. This should be discussed in the paper if necessary.

The authors express some considerations on the performance of the in-silico analyses respect to the experimental findings. Even ignoring the transferability issues I’ve raised above, it’s not surprising the methods aren’t completely accurate - but it should be noted that data from a single protein aren't enough to make general claims.

·

Basic reporting

The work presented here is based on a classic approach, but the authors did a remarkable job. One of the difficulties of in silico approaches is to see the limitations of approaches and analyses. Here, there are no such problems. The whole process is well brought into context.

Experimental design

The design is rigorous and well balanced.

Validity of the findings

The design is simple and clear, so the results are of similar quality. The conclusions are well stated.

Additional comments

A pleasant article to read. Well detailed and rigorous.

---

## Round 0.2 · Major Revisions

The Reviewers noted that, despite having some improvement, the ms still needs major revisions.

Please answer to all the concerns raised by the two Reviewers. Put a special attention into the correct validation of the mutations detected, in the accuracy of the references along the text (including in material and methods section when adequate),

Reviewer 1 ·

Basic reporting

Thanks to the authors for revising the paper and making an effort towards addressing my comments. Despite the paper has improved in different aspects, some critical key points still remain.

In terms of basic reporting, I feel the form of the article has improved – the methods section is now more clear and allowed me to understand much more of the experimental design. This withstanding there still are some mistakes in the article text that the authors should address through another round proofreading or by asking a native speaker to proofread it. The way the results and discussion are written is a bit misleading – sometimes it’s difficult to understand when the authors refer to their predictions and when they refer to experimental evidence.

The authors have added the raw data they got from their analyses – it’s in a machine-readable format even though not the easiest one (i.e. CSV or at least Excel files would be preferable) but it’s there.

Experimental design

I feel like the methods are much better described now

The authors use evidence from other organisms, even evolutionary distant from human, to validate the findings of their in silico approach in human. The effect of a mutation can’t be transferred that easily from organism to organism, for instance because of i) compensatory effects due to other changes in the sequence ii) compensatory effects in other proteins (for instance in surfaces that mediates binding) iii) the fact that related proteins don’t have necessarily perfectly superimposable function iv) the fact that there might be other compensatory mechanisms that mask effects of a loss-of-function mutation differently in different organisms. The opposite is also true - mutations that aren’t deleterious in other organisms might be deleterious in human.

The authors here i) derive a collection of mutations from different organisms, focusing on the most damaging mutations, ii) use bioinformatics predictors to assess the impact of these mutations in humans, iii) use evidence mostly from other organisms to validate their finding. The other studies that are linked by the authors follow a different approach: they derive potentially interesting SNPs from different sources, use bioinformatics predictors to understand which one could be the most damaging, collect evidence that such mutations can be deleterious in human. In the present work the approach is the opposite: predictions are used to validate findings from literature rather than the opposite.

Validity of the findings

Until the authors can’t clearly demonstrate this step, i.e. validate that the mutations they find are damaging in human or at least very sound reasons for which these mutations can be transferred, I feel like the research question remains too superficially explored.

Reviewer 3 ·

Basic reporting

There is an absence of literature references in the Results section.
There are numerous grammatical errors that need to be addressed.

Experimental design

This seems ok.

Validity of the findings

I am not sure of the validity of the findings or maybe that the validity of the findings are not currently stated. Whereas the prediction software does a good job of identifying mutations to conserved motifs within the Rad50 ABC ATPase and their effects on the core ATP binding and hydrolysis activity, other alterations of function are also noted which have not been described in the literature. To the best of my knowledge Rad50 is not a substrate for PKA or casein kinase II and it is not myristolylated. These predicted effects are probably false positives from the prediction and are being presented as valid results. Thus, the results and discussion need to be reworked to more clearly reflect what are validated and unvalidated findings from the prediction.

Additional comments

If I am understanding the “Analyses of Rad50 mutation deleterious effects” subsection of the Results, bioinformatics predicts that mutations in human Rad50 can have certain effects. For example, mutations in the Walker A affect catalytic and allosteric site, which has been shown. Do the authors next suggest that mutations in this region affect N-myristolylation, casein kinase II and protein kinase A phosphorylation of Rad50? Is Rad50 the substrate for these enzymes? Are this predictions biologically relevant? Similar statements are made about the coiled-coil domain and signature motif. It seems that some of the predictions are valid and others are not based on what is known about Rad50 structure/function.

The “Limitations of in silico predictions” subsection of the Discussion section needs to be greatly expanded to include more possible problems with the Results (see above). Again, I am not aware of any literature data that states that Rad50 is a substrate for PKA, casein kinase II, N-myrstolylation, etc. Too much weight is given to what is probably false positives from the prediction software.

In the Results section, the “Analysis of protein domains” subsection needs references. There are numerous crystal structures of Rad50 from archaea, bacteria, and a thermophilic eukarya. These features were also previously identified >19 years ago, and their role in Rad50 function is the subject of many of the mutations used in this study.
Generally, the Results section needs referencing.

The sentence on lines 412 and 413 is incorrect. The globular ABC ATPase head domain is formed by the N- and C-terminus. The Zn-hook is in the middle of the primary structure.

The Walker B mutation E1232Q (T. maritima E798Q) is ATPase dead. The analysis of this mutation in the Discussion (line 452-457) is incorrect. In general, more care has to be taken in describing the underlying effect of the mutations.

The sentence starting “Our analysis using PredictSNP have identified three mutations” on line 457 is confusing and maybe a bit misleading. CFTR is a member of the ABC ATPase superfamily like Rad50. Mutations in CFTR that result in some cases of CF have been made in Rad50 and analyzed as the references imply.

D1238N is a known mutation in Rad50 that has been observed in breast cancer. It effect has nothing to do with FHA binding. Studies by the Nelson lab utilizing T4 Rad50 have shown that it has problems with ATP hydrolysis.

On line 79, Carney at al reference is not formatted properly.
Line 96 should read “one of the most common types of genetic”.
Line 105 should read “the accelerating number of known SNPs have made it”.
Line 171, I am not sure what is meant by “mutations have been proceeded with pairwise alignment”.
Line 174 should read “mutations in human were manually refined,”.
Line 292 should read “literature pertaining to the topic was exhaustively”.
Line 322 should read “The regions with no 3D structure”.
Line 323 should read “which were mainly predicted to”.
Line 410 should read “Rad50 is a member of the”.
References in the Discussion are not consistently formatted.

---

## Round 0.3 · Major Revisions

The Reviewer noted some improvement in the writing but the ms still needs major revisions.

Please answer to all the concerns raised by the Reviewer, namely the transferability issue and how it affects the choice of mutations and the interpretation of results. Also adequate references to the interpretations of the findings are necessary in order to be able to re-consider your Ms.

Reviewer 1 ·

Basic reporting

Thanks to the authors for their resubmission.

Writing quality has improved from the last revision, even though the text isn't always easy to follow and contains mistakes. For instance (but not limited to) these sentences require attention:

171-172 "From this
172 analysis, we identified equivalent site of mutation in human. "
204 Prediction of deleterious effects of the identified Rad50 mutations were carried out
225 This is achieved by providing a general pathogenicity prediction
251 & Casadio, 2005) using default setting
Sequence homology search of the human Rad50 protein was performed against
267 NCBI nonredundant protein databases (equal and lesser 1E-05) (what's equal or lesser?)
342 All of these 42 mutations (based on human equivalent mutations) (these are human mutations derived from non-human ones)

Experimental design

I feel the authors haven't completely addressed the transferability issue and how it affects the choice of mutations and the interpretation of results - see below

Validity of the findings

The way in which the results are exposed is clearer but still a bit misleading, in that they are written in a way that suggests me that these are confirmed consolidated results while they are actually predictions. This is for instance the case in:

"mutations located at Walker A (Figure 2a) affect catalytic and allosteric site, loss or
gain of methylation, alteration of DNA binding, metal binding, ordered interface and the loss of relative solvent accessibility"
"These mutations were found to affect ATP binding site motif, N-myristolylation, casein kinase II (CK2), protein kinase A (PKA) phosphorylation site and Forkhead-associated (FHA) functional sites"
"These mutations were found to affect ATP binding site motif, N-myristolylation, casein kinase II (CK2), protein kinase A (PKA) phosphorylation site and Forkhead-associated (FHA) functional sites"
"We have also identified K6E, K132E and K105E mutations occurred at the coil-coiled domain or ATPase domain to be deleterious"
The authors should better cross-reference these findings with what is known in the literature (for instance they should at least provide evidence that binding/PTM sites they have predicted to be damaged by the mutations exist and are important)

The authors haven't really addressed my point on the transferability of the mutations they find and how this affects the selection of the mutations to be studied. This effect is also not considered in the interpretation of the results. This is especially evident for instance in the "Limitations of in silico prediction" section. While I agree that bioinformatics predictors aren't perfect, comparing prediction in human with experimental results in other organisms isn't really fair because of transferability issues (see my last revision). I don't think it's surprising that the predictors predict more damaging mutations in conserved sites, since they are typically important for function/structure and so mutations in them are more likely to be damaging. For less conserved regions, the effect in vivo is also likely to be more context-dependent (hence the transferability issue becomes even more important).

---

## Round 0.4 · accepted · Accept

The major concerns from the Reviewer have been addressed. However, the issues on transferability could be better integrated into the discussion. Please address this minor issue in the final version of the Ms.

Reviewer 1 ·

Basic reporting

I feel the reporting has improved to a point that is acceptable for publication

Experimental design

the reporting is clear enough that now predictions and experimental data are clearly distinguishable

Validity of the findings

The author have briefly acknowledged the issues on transferability, even though they are not well-integrated with the rest of the discussion. It's enough for a pass, even though I'd advise them to integrate them better in the discussion. I don't think this would be worth of another round of revisions.